# Neck Circumference and Its Association with Cardiometabolic Risk Factors in Pediatric Population

**DOI:** 10.3390/medicina55050183

**Published:** 2019-05-21

**Authors:** Carlos Adrián González-Cortés, Margarita Téran-García, Claudia Luevano-Contreras, Diana Patricia Portales-Pérez, Juan Manuel Vargas-Morales, Ana Cristina Cubillas-Tejeda, Patricia Elizabeth Cossío-Torres, Celia Aradillas-García

**Affiliations:** 1Centro de Investigacion Aplicada en Ambiente y Salud (CIACYT), Universidad Autonoma de San Luis Potosi (UASLP), San Luis Potosi 78210, Mexico; carlos.gzcortes@gmail.com; 2Human Development and Family Studies, Division of Nutritional Sciences, Cooperative Extension, Carle-Illinois College of Medicine, University of Illinois at Urbana-Champaign, Urbana, IL 61801, USA; teranmd@illinois.edu; 3Medical Sciences Department, University of Guanajuato, Leon Guanajuato 37320, Mexico; c.luevanocontreras@ugto.mx; 4Facultad de Ciencias Quimicas, Universidad Autonoma de San Luis Potosi, San Luis Potosi 78210, Mexico; dportale@uaslp.mx (D.P.P.-P.); lacrimas0@yahoo.com.mx (J.M.V.-M.); acris@uaslp.mx (A.C.C.-T.); 5Departamento de Salud Publica, Facultad de Medicina, Universidad Autonoma de San Luis Potosi, San Luis Potosi 78210, Mexico; patriciacossiotorres@yahoo.com.mx; 6Centro de Investigacion Aplicada en Ambiente y Salud, CIACYT/Facultad de Medicina, Universidad Autonoma de San Luis Potosi, San Luis Potosi 78210, Mexico

**Keywords:** neck circumference, anthropometric measurements, cardiometabolic risk, children

## Abstract

*Background and objectives*: To identify the relationship between neck circumference (NC) and cardiometabolic risk factors in children. *Materials and Methods*: Children and adolescents 6–18 years old (n = 548) from five counties of San Luis Potosí, México were included. Data was collected for biological markers (glucose and lipid profile) and anthropometric and clinical measurements—weight, height, NC, waist circumference (WC), and blood pressure (BP). Body mass index (BMI) was calculated using Quetelet formula (kg/m^2^). Descriptive analysis, correlation tests, and receiver operating characteristic (ROC) analysis were performed. *Results*: NC was highly correlated with BMI and WC in both genders (*p* <0.0001). The most frequent risk factor was high BMI (38.7%). Sensitivity and specificity analysis of NC and high BMI showed an area under the ROC curve of 0.887. *Conclusions*: According to our findings, NC is a simple, low-cost, and non-invasive measurement, which has a high association with high BMI and increased WC.

## 1. Introduction

Overweight and obesity are defined as an excessive accumulation of body adipocytes, which has many health implications, resulting in a reduced life expectancy and an increased risk of developing diabetes mellitus, cardiovascular disease, and some types of cancer [1]. The increased prevalence of obesity in the pediatric and adult population is a cause of alarm in healthcare systems globally [2].

Fat is distributed in different anatomical places and varies from 5–60% of the total body weight. Types of body fat storages include visceral adipose tissue (VAT) and subcutaneous adipose tissue (SAT). VAT represents 10% to 20% of the total body fat in men and 5% to 10% of the total body fat in women. SAT represents more than 80% of the body fat, and the most studied fat storage sites are abdominal, gluteus and femoral tissue [3]. Some studies have demonstrated the health implications of an imbalance in body fat distribution: the increase in both fat storages is associated with risk of metabolic diseases; especially the increase of VAT, which confers a higher risk [3,4,5].

Several direct and indirect techniques are available to measure adipose tissue distribution. Most of the direct methods are invasive and expensive—dual-energy X-ray absorptiometry (DEXA) and doubly labeled water (DLW)—and some of the indirect methods, including weight, height, and neck and waist circumferences, are difficult to replicate by untrained personnel. However, the advantages of these anthropometrical measurements are that they provide nutritional status information at low cost and can be practically implemented by clinicians [6].

In this regard, neck circumference (NC) is a practical anthropometric measurement used as an indicator of local fat storage in the upper body. It does not require multiple measurements for accuracy and is not influenced by environmental conditions (including the time of day or season) [7,8]. This anthropometric measure has optimal outcomes in the pediatric evaluation and can be used in clinical practice or epidemiological studies as a central obesity indicator [8]. Furthermore, NC has been associated with other chronic disease risk factors [9], such as high blood pressure and metabolic syndrome [10].

NC is positively correlated with other anthropometric measurements (body mass index (BMI) and waist circumference) and can be used as an independent cardiometabolic risk indicator in adults [11,12,13]. Accordingly, standard NC values in children are needed by age and gender to improve the translation of this measure to the clinical practice [14].

The NC has shown benefits over waist circumference (WC), for instance: NC does not require clothing removal (which is important in climate variations and customs of some cultures), is not altered by respiratory movements, and has demonstrated adequate inter- and intra-rater reliability; therefore, it does not require multiple measurements [15,16]. Also, NC has been associated with adiposity and obesity, which makes it a practical measure for large epidemiological studies, as well as for daily clinical practice [17]. Therefore, the objective of this study was to evaluate the relationship between NC and cardiometabolic risk factors in children.

## 2. Materials and Methods

### 2.1. Subjects

Participants were selected from public schools in the metropolitan areas of Ahualulco, Moctezuma, Villa de Guadalupe, and Matehuala, which are counties of a central state in Mexico: San Luis Potosí. Participants were students from elementary school, junior high school, and high school, from 6–18 years old, who studied during the 2015–2017 school years.

The sample was a stratified random sampling of each education level (elementary school, junior high school, and high school). First, for each education level, two schools were randomly selected from the complete list of the state of San Luis Potosí. Half of the schools were located in places with very high and very low marginalization index. The index of marginalization measures ten socioeconomic indicators (deficit variables). The final sample resulted in eight schools and 801 students accepted to participate and sign an inform consent; however, only 548 had all the complete data required and were included for final analysis. In addition, the sample size was calculated for this study using a 97% confidence interval and an acceptable margin of error of 5%, and the final sample size was 455 participants.

The data collected included: (1) a blood sample following an overnight fast to measure biological markers; and (2) anthropometric measurements, including height, weight, waist and neck circumference, and blood pressure.

Interested participants and their parents were informed about study goals and their rights for the project before signing the informed consent form. The study was performed according to the Helsinki Declaration and was reviewed and approved by the State Committee of Ethics in Health Research of San Luis Potosí, SLP/003-2015. All the participants received results of their tests, an explanation of their data, and recommend

### 2.2. Participant Selection

Participants for this observational and cross-sectional study were selected based upon the degree of completion of data available from the anthropometric measurements and blood sample. Only individuals with complete information for the outcome risk factors were included (n = 548). The participants were divided into two groups for later analysis: 6–11 years old and 12–18 years old.

### 2.3. Anthropometric Measures

All measures were conducted by trained and standardized health professionals (Nutritionists, Chemists, and Phlebotomists from the UASLP).

Height was measured using a mobile stadiometer SECA^®^ 213, 205 (Seca 213, Seca, Hanover, MD, USA, 2009) placed on a flat surface at a 90° angle, and was recorded to the nearest 0.1 cm. A Frankfort plane was defined as the imaginary line which passes through the lower edge of the orbit of the eye and the highest point of the external auditory canal [18].

Weight was collected on a calibrated electronic weighing scale TANITA^®^ UM-081 (Tanita UM-081, Tokyo, Japan) placed on a flat surface at a 90° angle, and recorded to the nearest 0.1 kg. BMI was calculated by Quetelet formula (weight/height^2^: kg/m^2^), and it was evaluated with Z scores of BMI for age (BMI/A), taking into consideration the date of birth and gender. BMI/A was classified according to the World Health Organization and using the WHO Anthro plus program [16]. There were six categories: (1) Severely wasted (below−3 Z -score); (2) Wasted (below −2 Z-score); (3) Normal (below −1 to 0 Z-score); (4) Possible risk of overweight (above 1 Z-score); (5) Overweight (above 2 Z-score), and 6) Obese (above 3 Z-score). High BMI for cardiometabolic risk was determined as those above 1 Z-score [19].

Waist circumference was measured using a LUFKIN^®^ Executive Thin line 2 m, W606PM metal tape (Lufkin W606PM, Sparks, MD, USA), to the nearest 0.1 cm, at the midpoint between the iliac crest and the last rib, taken in the narrowest part of the abdomen and at the end of normal expiration, with the subject in the standing position. Waist circumference risk level for cardiometabolic risk was determined as those above the >90th percentile, according to percentiles in Mexican-American children [20].

NC was measured at the level of the thyroid cartilage, using a LUFKIN^®^ Executive Thin line 2 m, W606PM metal tape (Lufkin W606PM, Sparks, MD, USA), aligned horizontally, with the children standing upright, head held erect, and eyes facing forward. The measurement was approximated to the nearest 0.1 cm.

### 2.4. Blood Pressure

Trained health care providers measured blood pressure (BP) according to “National High Blood Pressure Education Program Working Group on High Blood Pressure in Children and Adolescents.” BP was taken on the dominant arm in the seated position using appropriately sized Welch Allyn cuffs. According to the Pocket Guide to Blood Pressure Measurement in Children, elevated blood pressure was defined as >90th percentile [21].

### 2.5. Biological Markers

Fasting blood samples were collected by trained health professionals from the UASLP, after 8 h of minimal fasting. All samples were measured and analyzed in the Chemistry Faculty’s laboratory from the UASLP according to their internal protocols. Fasting plasma glucose, triglycerides, and HDL-cholesterol (HDL-C) were determined using the spectrophotometry method by A 15 equipment of Biosystems^®^ (Biosystems A 15, Barcelona, Spain).

High fasting glucose (GLUC) was defined as having GLUC >100 mg/dL. Serum triglycerides (TG) were determined as high as TG: 0–9 years >100 mg/dL and 10–19 years >130 mg/dL. Low serum HDL-C was defined as HDL-C <40 mg/D [22].

### 2.6. Statistical Analysis

The sample was divided into two groups, one with 6–11 years old children, and the other with 12–18 years old adolescents, to compare the variables. Descriptive statistics were performed for all variables, including mean values and standard deviations for continuous variables. Normality of distribution was checked for continuous variables using the Kolmogorov-Smirnov test. Student’s *t*-test was used to evaluate gender differences in continuous variables. X^2^ test was used to evaluate the difference of proportions in categorical variables.

Correlations between cardiometabolic risk factors and anthropometric variables were evaluated using Pearson’s correlation coefficient according to the normality of distributions, as males and females combined and males and females separately. A receiver operating characteristic (ROC) curve was constructed using values of NC and BMI related to categories for overweight/obesity according to WHO (WHO, 2008). The area under the ROC curve (AUC) was calculated using a 95% confidence interval (CI). *p*-values <0.05 were considered statistically significant. The statistical analysis was performed with IBM SPSS Statistics^®^ version 22 (SPSS, Chicago, IL, USA).

## 3. Results

A total of 548 children and adolescents were included for the final analysis, of which 286 (52.1%) were girls. Anthropometric and metabolic characteristics of the study population by gender are presented in Table 1. Also, an analysis of the divided sample by age (6–11 years old and 12–18 years old) was performed by gender (Table 2).

There were gender differences—NC, systolic blood pressure, glucose, and HDL-C mean values were significantly higher for boys. For girls, BMI and triglycerides mean values were significantly higher. Detailed information is presented in Table 1.

In the same way, we observed gender differences in the age groups: for the participants of the 6–11 years old group, glucose and HDL-C mean values were significantly higher for boys. For girls, triglycerides mean values were significantly higher. For participants of the 12–18 years old group, height, NC, and glucose mean values were significantly higher in boys (Table 2).

The prevalence of cardiometabolic individual risk factors indicates that high BMI (19.9%) and elevated blood pressure (19.8%) were the most common individual risk factors in boys and girls, respectively (Table 3).

Correlations between NC and cardiometabolic risk factors are presented in Table 4 for the total sample by gender. There was a strong and positive correlation of NC with BMI and WC in both genders. However, there was no correlation of NC with diastolic blood pressure in boys and glucose in girls.

Results of the sensitivity and specificity analysis of NC and BMI are presented as a receiver operator characteristic curve in Figure 1. The area under the ROC curve (AUROC) was 0.887, suggesting NC is useful in predicting overweight and obesity in children. For example, for this sample, a sensitivity of 90.2% and specificity of 75.9% predict BMI above the 85th percentile (overweight and obesity).

## 4. Discussion

The aim of this research was to identify the relationship between NC and cardiometabolic risk factors in children. We found that NC was significantly associated with BMI and WC. Likewise, high BMI and elevated blood pressure were the most common individual risk factors among our study sample.

Regarding the associations of NC and cardiometabolic risk factors, one study in Turkey evaluated the correlation between NC and cardio metabolic factors in 581 children and adolescents aged 5–18 years and found a strong association with BMI and WC in boys and girls [23]. Our results are also in agreement with research by Cassia et al. in Brazil, where a positive and high correlation between NC and BMI and WC was reported. This study also identified the relationship between NC and the components of metabolic syndrome in 388 children and adolescents aged 10–19 years in Brazil [24]. Unlike Cassia et al., in our study the participants recruited were from different schools from the state of San Luis Potosí, generating heterogeneity for the sample size and the location.

In a recent study, the high correlation between the WC, BMI, and NC has also been demonstrated in a population of Mexican children aged 6–11 years, showing similar values to our study in boys and girls [15].

Moreover, several studies have reported not only associations with anthropometrical indicators, but also with clinical and biological markers. More specifically, an association between NC and elevated blood pressure [10] and between NC and the homeostasis model assessment of insulin resistance (HOMA-IR) [25,26].

In accordance with our results, in 2016 Chunming Ma et al. performed a meta-analysis of six different studies, with the objective to evaluate the performance of NC for the assessment of overweight and obesity. They identified an AUROC of 0.8709 and showed moderate sensitivity and specificity in identifying overweight and obesity in children and adolescents [27]. Other authors have shown similar results, such as Coutinho et al. [28] and Gomez-Arbelaez et al. [29] in South American schoolchildren. Combining these results with ours, support is shown for the idea that NC is closely related with and capable of identifying upper-body adiposity [7,8].

Overweight/obesity and elevated blood pressure remain the most common risk factors among our study population. Others researchers have observed similar trends. For example, in American children (n = 1058), 6–18 years old, 37.7% were overweight/obese and 29.9% had elevated blood pressure, which was similar to our findings. In 2013, a study performed in the US evaluated 1058 children 6–18 years old, and it showed that overweight/obesity (37.7%) and elevated BP (29.9%) are highly prevalent, which is in line with our findings [30].

In Mexico, overweight and obesity are the leading health risks in the children’s population. Similar to our results, the National Health and Nutritional Survey 2016 (ENSANUT in Spanish) reported a prevalence of 33.2% and 36.3% of overweight and obesity in children and adolescents, respectively [31].

Our results suggest that NC is capable of identifying health risks and components of cardiometabolic risk factors in children. Since Mexico has a high prevalence of overweight and obesity, a practical anthropometric measure such as NC for early identification of population at risk is essential.

A limitation of this study is that anthropometric measurements were only measured once because of limited time to collect data in schools. For future research, it is important to assess body fat composition.

## 5. Conclusions

In conclusion, according to our findings NC is a simple, low-cost, and non-invasive measurement, which has a relationship with high BMI and high WC. Identification of children with cardiometabolic risk factors in early stages is essential to prevent future complications in their health and development. Since there is no reference to NC cut-off points in Mexican children, it is necessary to establish NC cut-off points by age and gender, standardized for Mexican children by region, to identify at-risk individuals.

## Figures and Tables

**Figure 1 medicina-55-00183-f001:**
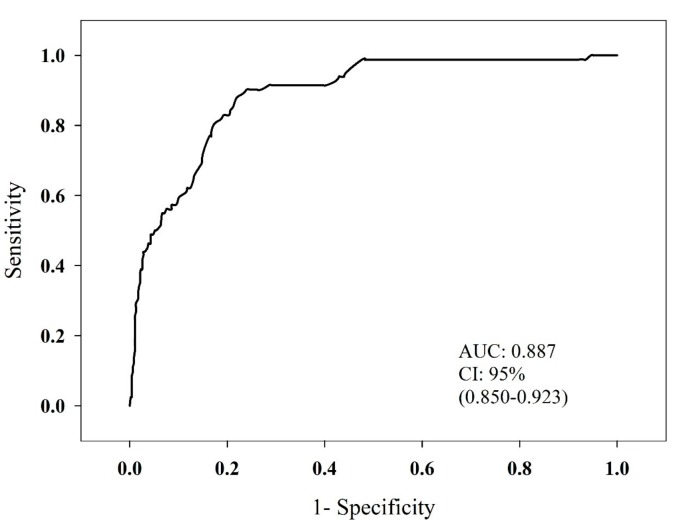
Receiver operating characteristic (ROC) curve for determining the optimal NC cut-off values for identifying high BMI, diagnosed by WHO criteria for BMI by age in children. AUC: area under curve, CI: confidence interval.

**Table 1 medicina-55-00183-t001:** Anthropometric and metabolic characteristics of the study population by gender.

Variable	Males (*n* = 262)	Females (*n* = 286)	*p*-Value
Weight (kg)	44.14 ± 17.86	44.59 ± 15.89	0.334 ^a^
Height (cm)	148.46 ± 17.12	146.12 ± 13.89	0.080 ^a^
BMI (kg/m^2^)	19.68 ± 4.62	20.60 ± 4.70	0.006 ^a^
WC (cm)	68.99 ± 13.72	70.06 ± 12.83	0.191 ^a^
NC (cm)	32.28 ± 4.68	30.87 ± 3.29	0.002 ^a^
SBP (mmHg)	114.64 ± 14.86	111.28 ± 11.94	0.000 ^b^
DBP (mmHg)	63.23 ± 9.88	63.27 ± 9.25	0.766 ^b^
GLUC (mg/dL)	89.65 ± 8.52	86.15 ± 8.01	0.000 ^a^
HDL-C (mg/dL)	55.95 ± 10.30	52.86 ± 10.75	0.000 ^a^
TG (mg/dL)	88.99 ± 53.21	101.69 ± 49.51	0.000 ^a^

BMI: body mass index; WC: waist circumference and NC: neck circumference. SBP: systolic blood pressure; DBP: diastolic blood pressure. GLUC: glucose; HDL-C: high-density lipoprotein cholesterol and TG: triglycerides. Values are presented as means ± SD. ^a^ U Mann-Whitney test for gender differences *p* < 0.05. ^b^
*t*-test for gender difference; *p* < 0.05.

**Table 2 medicina-55-00183-t002:** Anthropometric and metabolic characteristics of the study population by group age and gender.

Variable	6–11 Years n = 262		*p*-Value	12–18 Years n = 286		*p*-Value
Gender	Males n = 132 (50.4%)	Females n = 130 (49.6%)		Males n = 130 (23.5%)	Females n = 156 (29%)	
**Weight (kg)**	33.77 ± 11.62	35.10 ± 12.27	0.391 ^a^	54.50 ± 16.90	52.44 ± 14.19	0.398 ^a^
**Height (cm)**	135.04 ± 10.74	135.09 ± 12.49	0.876 ^a^	162.07 ± 10.04	155.24 ± 06.33	0.000 ^a^
**BMI (kg/m^2^)**	18.13 ± 4.11	18.69 ± 3.91	0.218 ^a^	21.27 ± 4.55	22.18 ± 4.71	0.058 ^a^
**WC (cm)**	63.40 ± 12.51	64.41 ± 11.69	0.424 ^a^	74.66 ± 12.43	74.73 ± 11.84	0.881 ^a^
**NC (cm)**	29.65 ± 4.03	29.05 ± 3.00	0.350 ^a^	34.95 ± 3.63	32.38 ± 2.71	0.000 ^a^
**SBP (mmHg)**	107.68 ± 12.43	106.32 ± 12.33	0.378 ^b^	121.71 ± 13.77	115.38 ± 9.90	0.106 ^b^
**DBP (mmHg)**	60.14 ± 10.16	60.31 ± 8.66	0.884 ^b^	66.37 ± 8.55	66.09 ± 8.93	0.120 ^b^
**GLUC (mg/dL)**	89.56 ± 7.54	86.95 ± 7.55	0.003 ^a^	89.73 ± 9.43	85.48 ± 8.34	0.000 ^a^
**HDL-C (mg/dL)**	56.70 ± 10.72	51.21 ± 9.38	0.000 ^a^	55.35 ± 10.25	54.29 ± 11.61	0.291 ^a^
**TG (mg/dL)**	83.07 ± 41.29	102.70 ± 53.46	0.001 ^a^	95.37 ± 65.27	100.46 ± 46.21	0.135 ^a^

BMI: body mass index; WC: waist circumference and NC: neck circumference. SBP: systolic blood pressure; DBP: diastolic blood pressure. GLUC: glucose; HDL-C: high-density lipoprotein cholesterol and TG: triglycerides. Values are presented as means ± SD. ^a^ U Mann-Whitney test for gender differences *p* < 0.05. ^b^
*t*-test for gender differences *p* < 0.05.

**Table 3 medicina-55-00183-t003:** Prevalence of cardiometabolic individual risk factors in children by age group.

Risk Factor	All (n = 548)	6–11 Years (n = 262)	12–18 Years (n = 286)	*p*-Value *
**Increased BMI**	38.3%	19.9%	18.4%	0.213
**Increased WC**	13.5%	8.6%	4.9%	0.003
**Elevated BP**	36.5%	16.7%	19.8%	0.485
**High TG**	23.0%	12.4%	10.6%	0.115
**High Fasting Glucose**	8.0%	3.1%	4.9%	0.204
**Low HDL-C**	6.9%	3.8%	3.1%	0.340

BMI: body mass index; WC: waist circumference. BP: blood pressure. TG: triglycerides. HDL-C: high-density lipoprotein cholesterol * X^2^ test for age group differences *p* < 0.05.

**Table 4 medicina-55-00183-t004:** Comparison between neck circumference and cardiometabolic risk factors using Pearson’s correlation coefficient.

FEMALES (n = 286)								
	*NC*	*BMI*	*WC*	HDL-C	*TG*	*SBP*	*DBP*	*GLUC*
*NC*	- -	0.78 **	0.79 **	−0.31 **	0.31 **	0.33 **	0.13 *	0.04
*BMI*	0.81 **	- -	0.88 **	−0.28 **	0.32 **	0.22 *	0.08	0.04
*WC*	0.81 **	0.92 **	- -	−0.35 **	0.36 **	0.27 **	0.11 *	0.02
*HDL-C*	−0.28 **	−0.30 **	−0.32 **	- -	−0.23 **	−0.03	0.01	- -
*TG*	0.25 **	0.31 **	0.34 **	−0.24 **	- -	0.20 *	0.09	0.07
*SBP*	0.27 **	0.25 **	0.25 **	−0.05	0.15 *	- -	0.55 **	0.28 **
*DBP*	0.11	0.16 *	0.18 *	−0.06	0.14 *	0.50 **	- -	0.25 **
*GLUC*	0.12 *	0.15 *	0.14 *	−0.13	0.08	0.38 **	0.33 **	- -
**MALES (n = 262)**								

NC: neck circumference; BMI: body mass index; WC: waist circumference. HDL-C: high-density lipoprotein cholesterol and TG: triglycerides. SBP: systolic blood pressure; DBP: diastolic blood pressure. GLUC: glucose. * *p* < 0.05, ** *p* < 0.0001.

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
