# Peer review of "Neck Circumference and Its Association with Cardiometabolic Risk Factors in Pediatric Population"

_medicina, 2019, doi:10.3390/medicina55050183_

Round 1
Reviewer 1 Report
In this study, titled “Neck circumference as an alternative anthropometric measure and its relationship with the cardiometabolic risk factors in pediatric populations”, the authors analysed the correlation between neck circumference and cardiometabolic risk factors in a sample of children. I have found the topic very interesting and the manuscript well written. I have just some minor suggestions:
- The title is very long. I suggest to to shorten it
- Lines 89-90. Please check your description of the Frankfurt horizontal plane. The description and the anthropometric point are not correct.
- Line 101. Your description of waist circumference is not correct. Please check and correct. The waist circumference must be taken in the narrowest part of the abdomen, between the last rib and the iliac crest.
- In the results section you don’t need to state every time the p value because it is already reported in the tables.
- In table 4 you reported the values of females but not of males.
Author Response
Response to Reviewer 1 Comments
Below please find our point-by-point response to the reviewer comments.
The changes made in the manuscript are indicated by page, and lines. The new text is marked in yellow.
Comments and suggestions for authors:
In this study, titled: “Neck circumference as an alternative anthropometric measure and its relationship with the cardiometabolic risk factors in pediatric populations”, the authors analysed the correlation between neck circumference and cardiometabolic risk factors in a sample of childre. I have found the topic very interesting and the manuscript well written. I have just some minor suggestions:
1. The title is very long. I suggest to shorten it:
Response 1: Thank you very much for your suggestion. We changed the title as follows:
Neck circumference and its association with cardiometabolic risk factors in pediatric population.
2. Lines 89-90: Please check your description of the Frankfurt horizontal plane. The description and the anthropometric point are not correct.
Response 2: Thank you very much for your suggestion. We changed the description in page 3, lines 100, and 101 as follows:
A Frankfort plane was defined as the imaginary line which passes through the lower edge of the orbit of the eye and the highest point of the external auditory canal.
3. Line 101: Your description of waist circumference ir not correct. Please check and correct. The wais circumference must be taken in the narrowest part of the abdomen, between the last rib and the illiac crest.
Response 3: Thank you very much for your suggestion. We changed the description in page 3, lines 110-112 as follows:
Waist circumference was measured using a LUFKIN® Executive Thin line 2m, W606PM metal tape, to the nearest 2.5px at the midpoint between the iliac crest and the last rib, taken in the narrowest part of the abdomen, and at the end of normal expiration.
4. In the results section you don´t need to state every time the p value because it is already reported in the tables.
Response 4: Thank you very much for your suggestion. The p values in the results section were removed, only those reported in the tables were left.
5. In table 4 you reported the values of females but not of males.
Response 5: Thank you very much for your suggestion. We changed table 4, and now the values of the correlations for females and maled are included, female above and males below.

Reviewer 2 Report
The manuscript discusses the use of neck circumference in predicting cardio metabolic risk in the paediatric population. There is a need for further information on the advantages of using this method over similarly practical methods. Further comments are described below:
Line 53: The advantage of using neck circumference relies in its practicality and the fact that it is cheap and that it could be used in clinical practice. However, waist circumference measurement is a similarly practical and cheap method for assessing cardiometabolic risk. Therefore, you need to clearly specify the benefit of using neck circumference to assess this risk, compared to WC. The latter has been shown to be an independent risk factor for diabetes and cardiometabolic risk factors in children. Could NC, for instance, be used as an alternative to WC in culturally sensitive people?
Line 105: Have you measured NC more than once in each participant? Please specify.
Line 123: You need to justify your sample size. Please include power analysis.
Line 171: Same point as above. It seems NC and WC have resulted in similar correlations with blood pressure and biochemical markers, so what would be the advantage of the use of NC?
Line 195: Rephrase.
Line 185: The discussion is weak. You need to interlink ideas together and critically discuss your results. You also need to expand to include a ‘limitations’ section.
Author Response
Response to Reviewer 2 Comments
Below please find our point-by-point response to the reviewer comments.
The changes made in the manuscript are indicated by page, and lines. The new text is marked in yellow.
The manuscript discusses the use of neck circumference in predicting cardio metabolic risk in he paediatric population. There is a need for further information on the advantages of using this method over similarly practical methods. Further comments are described below:
1. Line 53: The advantage of using neck circumference relies in its practicality and the fact that is cheap and that it could be used in clinical practice. However, waist circumference measurements is a similarly practical and cheap method for assessing cardiometabolic risk. Therefore, you need to clearly specify the benefit of using neck circumference to assess this risk, compared to WC. The latter has been shown to be an indepent risk factor for diabetes and cardiometabolic risk factor in children. Could NC, for instance, be used as an alternative to WC in culturally sensitive people?
Response 1: Thank you very much for your suggestion. We included a paragraph in the introduction section, page 2, line 62-66 describing the benefits of NC over WC.
The NC has shown benefits over WC, for instance: NC does not require clothing removal (which is important in climate variations & customs of some cultures), it is not altered by respiratory movements, and it has demonstrated adequate inter- and intra-rater reliability; therefore, it does not require multiple measurements. Also, NC has been associated with adiposity and obesity which makes it a practical measure for large epidemiological studies as well as for daily clinical practice.
2. Line 105: Have you measured NC more tan once in each participant? Please specify.
Response 2: NC was measured only once in each participant and it is declared as one of the limitations of this sudy in page 7, line 236.
3. Line 123: You need to justify your sample size. Please include power analysis.
Response 3: Thank you very much for your suggestion. We included a paragraph in the Materials and Methods section, page 2, line 74-81 justifying sample size, that is the reason why the power analysis is not included.
4. Line 171: Same point as above. It seems NC and WC have resulted in similar correlations with blood pressure and biochemical markers, so what would be the advantage of the use of NC?
Response 4: Thank you for your observation. The disadvantage of using WC are the clothing removal in children, and anthropometric variations and alterations by postprandial abdominal distention which are overcome by NC. That is why we identified advantages of NC.
5. Line 195: Rephrase.
Response 5: Thank you for your observation. The line was re-written.
6. Line 185: The discussion is weak. You need to interlink ideas together and critically discuss your results. You also need to expand to include a “limitations” section.
Response 6: Thank you for your observation. The discussion section was re-written. Page 7, line 201-237.
Discussion
The aim of this research was to identify the relationship between NC and cardiometabolic risk factors in children. We found that NC was significantly associated with BMI and WC. Likewise, high BMI and elevated blood pressure were the most common individual risk factors among our study sample.
Regarding the associations of NC and cardiometabolic risk factors, one study in Turkey evaluated the correlation between NC and cardio metabolic factors in 581 children and adolescents aged 5-18 years and found a strong association with BMI and WC in boys and girls [23]. Our results are also in agreement with a research by Cassia et al. In Brazil, where a positive and high correlation between NC and BMI and WC was reported. This study also identified the relationship between NC and the components of metabolic syndrome in 388 children and adolescents aged 10-19 in Brazil [24]. Unlike Cassia et al., in our study the participants recruited were from different schools from all the state of San Luis Potosí, generating heterogeneity for the sample size and the location.
In a recent study, the high correlation between the WC, BMI, and NC has also been demonstrated in a population of Mexican children aged 6-11 years, showing similar values to our study in boys and girls [15].
Moreover, several studies have reported not only associations with anthropometrical indicators but also with clinical and biological markers. More specifically, an association between NC and elevated blood pressure [10]; and between NC and HOMA-IR [25, 26].
In accordance with our results, in 2016 Chunming Ma et al., performed a meta-analysis of 6 different studies with the objective to evaluate the performance of NC for the assessment of overweight and obesity. They identified an AUROC of 0.8709 and showed moderate sensitivity and specificity in identifying overweight and obesity in children and adolescents [27]. Other authors also have shown similar results such as Coutinho et al, [28] and Gomez-Arbelaez et al [29] in South American schoolchildren. Taking together these results along with ours support the idea that NC is closely related and capable of identifying upper-body adiposity [7-8].
Overweight/obesity and elevated blood pressure remain the most common risk factors among our study population. Others researchers have observed similar trends. For example, in American children (n=1058), 6-18 years old 37.7% were overweight/obesity, and 29.9% had elevated blood pressure, which was similar to our findings. In 2013, a study performed in the US evaluated 1058 children 6-18 years old, and it showed that overweight/obesity (37.7%) and elevated BP (29.9%) are highly prevalent, which is in line with our findings [30].
In Mexico, overweight and obesity are the leading health risk in children’s population. Similar to our results, the National Health and Nutritional Survey 2016 (ENSANUT in Spanish) reported a prevalence of 33.2% and 36.3% of overweight and obesity in children and adolescents respectively [31].
Our results suggest that NC is capable of identifying health risks and components of cardio-metabolic risk factors in children. Since Mexico has a high prevalence of overweight and obesity, a practical anthropometric measure such as NC for early identification of population at risk is essential.
A limitation of this study is that anthropometric measurements were only measured once because of limited time to collect data in schools. For future research, it is important to assess body fat composition.

Round 2
Reviewer 2 Report
The manuscript has improved and most changes have been addressed. Discussion is now better but remains weak. The sample size seemed to be calculated retrospectively, which is undesirable.